# Cost and cost effectiveness of reactive case detection (RACD), reactive focal mass drug administration (rfMDA) and reactive focal vector control (RAVC) to reduce malaria in the low endemic setting of Namibia: an analysis alongside a 2×2 factorial design cluster randomised controlled trial

Henry Ntuku ,[1] Cara Smith-Gueye,[1] Valerie Scott,[1] Joseph Njau,[2] Brooke Whittemore,[3] Brittany Zelman,[1] Munyaradzi Tambo,[4] Lisa M Prach,[1] Lindsey Wu,[5] Leah Schrubbe,[1] Mi-Suk Kang Dufour,[6] Agnes Mwilima,[7] Petrina Uusiku,[8] Hugh Sturrock,[1] Adam Bennett,[1] Jennifer Smith,[1] Immo Kleinschmidt,[9,10] Davis Mumbengegwi,[4] Roly Gosling,[1] Michelle Hsiang[1,3]

For numbered affiliations see end of article.

**Correspondence to**
Dr Henry Ntuku;
hntuku@path.org

## ABSTRACT

**Objectives** To estimate the cost and cost effectiveness of reactive case detection (RACD), reactive focal mass drug administration (rfMDA) and reactive focal vector control (RAVC) to reduce malaria in a low endemic setting.

**Setting** The study was part of a 2×2 factorial design cluster randomised controlled trial within the catchment area of 11 primary health facilities in Zambezi, Namibia.

**Participants** Cost and outcome data were collected from the trial, which included 8948 community members that received interventions due to their residence within 500 m of malaria index cases.

**Outcome measures** The primary outcome was incremental cost effectiveness ratio (ICER) per in incident case averted. ICER per prevalent case and per disability-adjusted life years (DALY) averted were secondary outcomes, as were per unit interventions costs and personnel time. Outcomes were compared as: (1) rfMDA versus RACD, (2) RAVC versus no RAVC and (3) rfMDA+RAVC versus RACD only.

**Results** rfMDA cost 1.1× more than RACD, and RAVC cost 1.7× more than no RAVC. Relative to RACD only, the cost of rfMDA+RAVC was double ($3082 vs $1553 per event). The ICERs for rfMDA versus RACD, RAVC versus no RAVC and rfMDA+RAVC versus RACD only were $114, $1472 and $842, per incident case averted, respectively. Using prevalent infections and DALYs as outcomes, trends were similar. The median personnel time to implement rfMDA was 20% lower than for RACD (30 vs 38 min per person). The median personnel time for RAVC was 34 min per structure sprayed.

**Conclusion** Implemented alone or in combination, rfMDA and RAVC were cost effective in reducing malaria

## STRENGTHS AND LIMITATIONS OF THIS STUDY

⇒ This is the first study to evaluate the costs and cost effectiveness of two innovative malaria transmission reducing strategies, reactive focal mass drug administration and reactive focal vector control, used alone and in combination.
⇒ Estimates are derived from prospective data collection of costs and robust measures of effectiveness provided by a rigorous community randomised controlled trial.
⇒ In a secondary analysis, the study interventions were also found to be cost effectiveness to avert disability-adjusted life years.
⇒ The study design did not allow comparison of each intervention alone, since half of the clusters in each study arm compared received another intervention.

incidence and prevalence despite higher implementation costs in the intervention compared with control arms. Compared with RACD, rfMDA was time saving. Cost and time requirements for the combined intervention could be decreased by implementing rfMDA and RAVC simultaneously by a single team.

**Trial registration number** NCT02610400; Post-results.

## INTRODUCTION

In recent years, progress in malaria control and elimination, using standard approaches, has faltered,[1] leading to plateauing case numbers and seasonal outbreaks.[2] New

malaria elimination approaches are needed, yet there are limited data on their costs and cost effectiveness. In a context of declining donor funding, insufficient human resources and competing health priorities, economic evidence regarding malaria elimination interventions is needed to guide decision-making.

In areas with declining transmission intensity, a large proportion of infections are asymptomatic and tend to cluster geographically and temporally.[3–5] Since malaria vectors are still present, these infections may seed further transmission in their immediate neighbourhood, leading to focal outbreaks or clusters of malaria cases.[6] To prevent wider epidemics, community-based targeted interventions in response to passively identified malaria cases are recommended and implemented by programmes aiming to interrupt transmission.[7 8] One widely implemented intervention is reactive case detection (RACD), whereby individuals residing near passively detected index cases within a predefined radius are screened with a malaria rapid diagnostic test (RDT) and treated if results are positive. However, RACD may have limited impact because RDTs fail to identify low-density parasite infections, which predominate in low transmission settings.[4 9–12] To circumvent this challenge of missed low-density infections, reactive focal mass drug administration (rfMDA) is an alternative approach in which the same household members and neighbours of index cases are presumptively treated with an antimalarial drug irrespective of infection status.[13] Reactive focal vector control (RAVC) is another promising elimination strategy, where focal indoor residual spraying (IRS) is targeted to households near index cases identified through passive case detection.[13] RAVC ensures sufficient IRS coverage in the areas at highest risk of malaria transmission. These households may have been insufficiently covered during a national IRS campaign, typically held months before the malaria season starts. The combination of rfMDA and RAVC has been shown to be more effective than the interventions administered alone.[13]

In addition to an intervention's effectiveness in reducing transmission, costs and cost effectiveness are key considerations in the decision-making process for National Malaria Control Programs and countries embarking on malaria elimination. Given that malaria control budgets are limited, interventions or their combinations must be chosen based on expected cost and impact. While a few studies have examined costs associated with RACD,[14 15] there is very little information available on economic evaluations of rfMDA. The limited data available report on the costs of focal mass drug administration following community-wide screening[16] or mass drug administration (MDA), which is typically administered as a mass campaign.[17] Compared with standard IRS, RAVC has been shown to be cost effective in a very low transmission setting.[18] To the best of our knowledge, there are no published economic data on RAVC, nor rfMDA+RAVC, in a low transmission setting.

This paper reports the findings of an economic evaluation of three reactive strategies—RACD, rfMDA and RAVC—carried out as part of a cluster randomised controlled trial (CRCT) conducted in Zambezi Region, Namibia, in 2017.[13] The goal of the trial was to investigate the effectiveness and feasibility of novel interventions that target the parasite reservoir in humans (rfMDA) and the vector (RAVC) and are applied in a reactive and focal fashion (eg, in response to confirmed, passively identified malaria cases). Costs of these interventions were compared singly and in combination to their respective current standard of care (eg, rfMDA vs RACD, RAVC vs no RAVC and rfMDA+RAVC vs RACD+no RAVC). Personnel time required for the interventions was also compared.

## METHODS

### Study site and description of the trial

The study was conducted from January 2017 to December 2017 in Zambezi region, northern Namibia. From 2010 to 2015, annual parasite incidence was low (<15/1000 population), likely due to an increase in malaria funding and scaling up of effective interventions, including improved case management and wide-scale implementation of preseason IRS campaigns with dichlorodiphenyltrichloroethane or deltamethrin.[19] However, there was a resurgence in the year prior to the trial (2016) and incidence rose to 33/1000 population. Transmission is almost entirely due to *Plasmodium falciparum*. *Anopheles arabiensis* is the dominant vector, with the high transmission season occurring from January to June.

As reported elsewhere, the parent study was a CRCT with 2×2 factorial design.[13 20] In brief, 56 census enumeration areas (EAs) within the catchment area of 11 health facilities were randomised to receive rfMDA or RACD, and RAVC or no RAVC, resulting in four study arms: (1) RACD only, (2) RACD+RAVC, (3) rfMDA only and (4) rfMDA+RAVC (table 1). The primary outcome was cumulative incidence of local malaria cases and the secondary outcome was infection prevalence as assessed in an endline survey. The total study population was 18 803 individuals. Study interventions were triggered by an index case diagnosed at the study health facilities by either RDT or microscopy and reported through an electronic rapid reporting system. Within approximately 2 weeks of reporting, a study team consisting of a field investigator, nurse and driver/data collector visited the index case household and neighbouring households within a 500-metre radius with a target of enrolling at least 25 individuals for RACD and rfMDA. For RAVC, a separate team consisting of a spray team leader/driver and a spray assistant visited the same location and sprayed the index case household and seven closest households. For each intervention, one follow-up visit was made if the enrolment target was not reached at the initial visit.

RACD consisted of testing for malaria using an RDT (CareStart Malaria HRP2/pLDH Pf/PAN, Access Bio, Somerset, New Jersey, USA) and treating those with positive results with artemether–lumefantrine (AL) (Coartem, Novartis, Switzerland) and single low dose primaquine

**Table 1** Main trial design and costing outcome measures

| | | Human reservoir | |
|---|---|---|---|
| | | **RACD** | **rfMDA** |
| | | ▶ 28 clusters | ▶ 28 clusters |
| | | ▶ Pop at risk: 9898 | ▶ Pop at risk: 8905 |
| | | ▶ No. of events: 178 | ▶ No. of events: 164 |
| | | ▶ No. of people: 4701 | ▶ No. of people: 4247 |
| | | ▶ No. of RDT identified infections: 114 | |
| Mosquito reservoir | **No RAVC** | **RACD only arm** | **rfMDA only arm** |
| | ▶ 28 clusters | ▶ 14 clusters | ▶ 14 clusters |
| | ▶ Pop at risk: 9339 | ▶ Pop at risk: 4742 | ▶ Pop at risk: 4597 |
| | ▶ No. of events: 170 | ▶ No. of events: 82 | ▶ No. of events: 88 |
| | ▶ No. of people: 4369 | ▶ No. of people: 2188 | ▶ No. of people: 2181 |
| | | ▶ No. of RDT identified infections: 52 | |
| | **RAVC** | **RACD+RAVC arm** | **rfMDA+RAVC arm** |
| | ▶ 28 clusters | ▶ 14 clusters | ▶ 14 clusters |
| | ▶ Pop at risk: 9464 | ▶ Pop at risk: 5156 | ▶ Pop at risk: 4308 |
| | ▶ No. of events: 172 | ▶ No. of events: 96 | ▶ No. of events: 76 |
| | ▶ No. of people: 4032 | ▶ No. of people: 2513 | ▶ No. of people: 2066 |
| | | ▶ No. of RDT identified infections: 62 | |

RACD, reactive case detection; RAVC, reactive focal vector control; RDT, rapid diagnostic test; rfMDA, reactive focal mass drug administration.

(Primaquine, Remedica, Cyprus) per national policy. Dried blood spots were collected for molecular analysis. For rfMDA, eligible individuals were offered a presumptive, weight-appropriate, 3-day treatment course of AL only (Coartem, Novartis, Switzerland). RAVC consisted of spraying ceilings and walls of sleeping structures with pirimiphos-methyl (Actellic 300 CS, Syngenta AG, Basel, Switzerland) per standard procedures.[21] To assess medication adherence, the team performed a follow-up pill count 7–10 days after the intervention for all RACD participants who were prescribed AL and among a sample of rfMDA participants.

The results of the trial showed that rfMDA and RAVC implemented alone and in combination reduced malaria incidence and parasite prevalence. Specifically, rfMDA and RAVC each reduced malaria incidence by nearly 50% when compared with their respective controls, RACD and no RAVC, and their combination reduced malaria incidence by 75% compared with RACD only. Similarly, rfMDA and RAVC reduced prevalence by 41% and 64%, compared with their respective controls RACD and no RAVC and their combination reduced prevalence by 84% compared with RACD only.[13]

### Collection of costing data

The costing analysis was undertaken from the provider or programme perspective. An ingredients-based or microcosting approach was used for line-item cost estimation. Inputs were identified, quantified, valued and classified into activity category.[22] Cost data were collected retrospectively from the financial expenditure records. No adjustments for inflation were made. Costs of items were valued according to their market values in the year when they were purchased, in either Namibia dollars (NAD) or

US$. Costs in NAD were converted to US$ using the official 2017 average exchange rate of US$1 per NAD14.28.[23] Data were entered into a costing tool in Microsoft Excel 2010.

Costs were classified as either capital or recurrent, showing the difference between investment once-off costs versus those that represent the running costs of ongoing programme implementation. Capital costs included vehicles, computers, tablets, printers, photocopiers and office furniture. For capital items such as vehicles, office furniture and computers estimated to have life years beyond the study period, a discount rate of 3% and depreciation were applied before including 1-year value of the equipment to the costs of the study (estimated project evaluation duration).[24] Recurrent costs were grouped into one of four categories: consumables, services, personnel or training and meetings. Consumables included field supplies and commodities related to the study intervention (eg, RDTs, drugs and insecticides). Services included fuel for vehicles, airtime for staff, data bundles for tablets, maintenance of equipment and office rental. Personnel costs included salaries and fringe benefits at a partner pay scale. Based on the time and motion observation (see below), a percentage of the personnel time spent on each intervention was applied to the full annual costs (including salary, benefits and any types of allowances or per diem). Costs were assigned to either RACD, rfMDA or RAVC. For items shared by different intervention types, a percentage of use was allocated based on trial implementation data. The costing assessment was limited to the basic intervention level to resemble programmatic costs as close as possible. Costs related to research (ie, international collaboration, the endline survey or molecular analyses) were excluded.

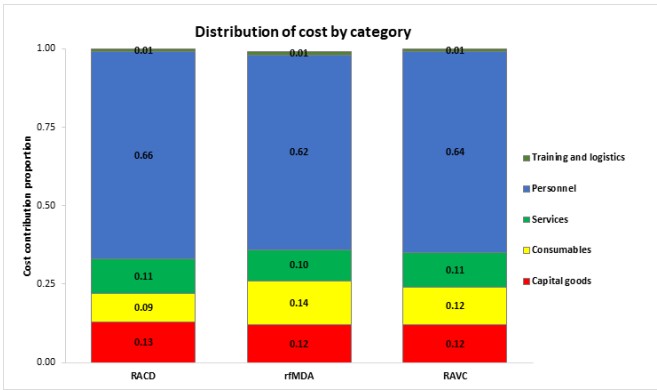

**Figure 1** Cost category breakdown for RACD, rfMDA and RAVC. RACD, reactive case detection; RAVC, reactive focal vector control; rfMDA, reactive focal mass drug administration.

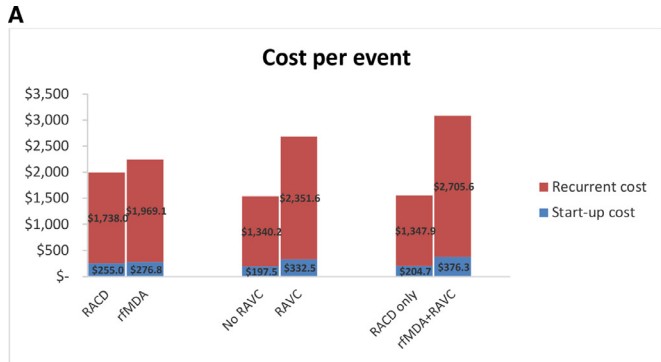

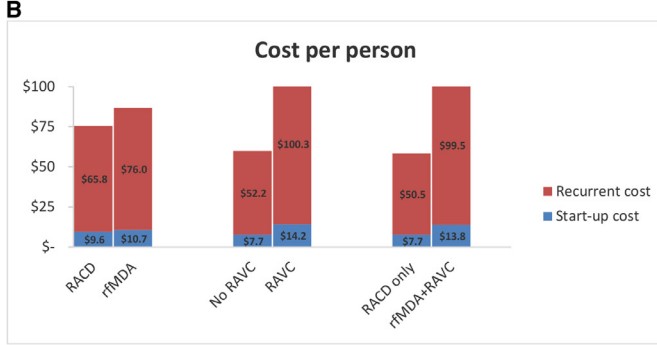

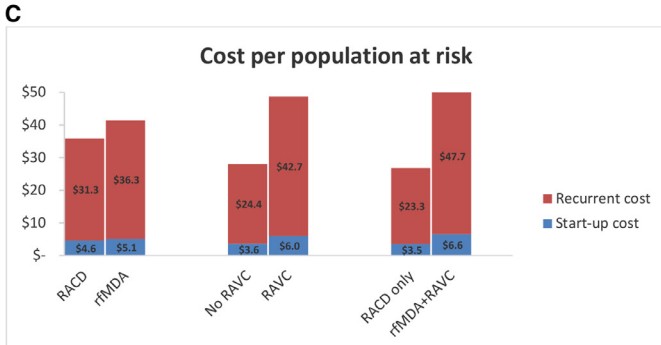

**Figure 2** Start-up and recurrent costs by comparison groups for cost per event (A), cost per person (B) and cost per PAR (C). PAR, population at risk; RACD, reactive case detection; RAVC, reactive focal vector control; rfMDA, reactive focal mass drug administration.

### Personnel time

To assess the average time spent to complete the different interventions, a time and motion approach was used.[25] A field coordinator observed field personnel participating in the event, and recorded time spent by each staff member performing specific tasks. The sum of time spent by each personnel on each task was used to compute the total average time to complete the event. Data on personnel time were collected as part of supervisory visits for a sample of 10 events for each strategy (rfMDA, RACD and RAVC). Observed activities included: preparation time in the office, driving time and event start and end times at the village. Data regarding the number of people treated (rfMDA) or tested and treated if positive (RACD) or the total number of structures sprayed (RAVC) were obtained from the trial database.

### Collection of outcome data

Data on population at risk (PAR) by study arms (defined as the total population in the study arm during the study period), number of events conducted, number of households sprayed and the total population residing in the sprayed households, number of people tested or presumptively treated, number of RDT-detected infections during RACD and malaria case incidence and infection prevalence data were retrieved from the trial database[13] (table 1 and online supplemental appendix 1).

### Data analysis

The following three comparison groups were used to compare intervention to control arms and evaluate the cost and cost effectiveness of interventions targeting the parasite reservoir: in humans, in mosquitoes and in both humans and mosquitoes, respectively (online supplemental appendix 2): (1) rfMDA vs RACD, (2) RAVC versus no RAVC and (3) rfMDA+RAVC versus RACD only. Costing data reflect the way that interventions were implemented in the 2×2 factorial design. Specifically, for the rfMDA versus RACD comparison, expenses for the rfMDA group included those from the rfMDA only and rfMDA+RAVC arms, and expenses for the RACD group included those from the RACD only and RACD+RAVC arms. The same approach was used for the RAVC versus no RAVC comparison, in which all expenses occurring in the RAVC arm (RACD+RAVC and rfMDA+RAVC) were compared with costs occurring in the no RAVC arm (RACD only and rfMDA only). With this approach, the no RAVC arm had non-zero costs and could be compared with RAVC. For the rfMDA+RAVC versus RACD only, the total costs of the combined interventions were compared with the total costs of RACD only.

To compare the costs, the following cost outcomes indicators were calculated: (1) total cost of the intervention for the duration of the 1-year trial, (2) cost per intervention event, calculated as the total cost of the strategy divided by the total number of responses conducted, (3) cost per individual, calculated as the total cost of the strategy divided by either the total number of persons

**Table 2** Total and unit costs per comparison group (cost in 2017 in US$)

| Comparison groups | Average costs (start-up and recurrent) | | | | | Recurrent costs only | | | | |
|---|---|---|---|---|---|---|---|---|---|---|
| | Total costs | Cost per event | Cost per person | Cost per PAR | Cost per infection identified | Total costs | Cost per event | Cost per person | Cost per PAR | Cost per infection identified |
| Human reservoir | | | | | | | | | | |
| RACD | $354 750 | $1993 | $75.5 | $35.8 | $3112 | $309 368 | $1738 | $65.8 | $31.3 | $2714 |
| rfMDA | $368 321 | $2246 | $86.7 | $41.4 | – | $322 940 | $1969 | $76.1 | $36.3 | – |
| Mosquito reservoir | | | | | | | | | | |
| No RAVC | $261 409 | $1537 | $59.8 | $28.0 | – | $227 837 | $1340 | $52.2 | $24.4 | – |
| RAVC | $461 661 | $2684 | $114.5 | $48.8 | – | $404 470 | $2352 | $100.3 | $42.7 | – |
| Human and mosquito reservoir | | | | | | | | | | |
| RACD only | $127 312 | $1553 | $58.2 | $26.9 | $2448 | $110 526 | $1348 | $50.5 | $23.3 | $2126 |
| rfMDA+RAVC | $234 223 | $3082 | $113.4 | $54.4 | – | $205 628 | $2706 | $99.5 | $47.7 | – |

PAR, population at risk; RACD, reactive case detection; RAVC, reactive focal vector control; rfMDA, reactive focal mass drug administration.

**Table 3** Impact indicators and ICER per comparison group (cost in 2017 in US$)

| Comparison groups | Predicted incidence per 1000 person-years (95% CI) | No. of incident cases | No. of incident cases averted | DALYs averted | Predicted infection prevalence (95% CI) | No. of prevalent infections | No. of prevalent infections averted | ICER per incident case averted | ICER per prevalent infection averted | ICER per DALY averted |
|---|---|---|---|---|---|---|---|---|---|---|
| Human reservoir | | | | | | | | | | |
| RACD | 42.1 (30.0 to 54.2) | 396 | Ref. | Ref. | 3.3% (2.7 to 4.0) | 310 | Ref. | Ref. | Ref. | Ref. |
| rfMDA | 29.5 (19.0 to 40.0) | 277 | 119 | 140 | 2.4% (1.6 to 3.2) | 226 | 84 | $114 | $162 | $97 |
| Mosquito reservoir | | | | | | | | | | |
| No RAVC | 43.1 (28.7 to 57.6) | 405 | Ref. | Ref. | 3.5% (2.7 to 4.3) | 329 | Ref. | Ref. | Ref. | Ref. |
| RAVC | 28.6 (21.7 to 35.4) | 269 | 136 | 160 | 2.7% (1.8 to 3.3) | 254 | 75 | $1472 | $2670 | $1248 |
| Human and mosquito reservoir* | | | | | | | | | | |
| RACD only | 52.5 (30.4 to 74.6) | 238 | Ref. | Ref. | 3.1% (2.0 to 4.1) | 140 | Ref. | Ref. | Ref. | Ref. |
| rfMDA +RAVC | 24.6 (14.7 to 34.5) | 111 | 127 | 150 | 1.8% (0.5 to 2.9) | 81 | 59 | $842 | $1812 | $714 |

*Based on half of the clusters (14).
DALY, disability-adjusted life year; ICER, incremental cost effectiveness ratio; RACD, reactive case detection; RAVC, reactive focal vector control; rfMDA, reactive focal mass drug administration.

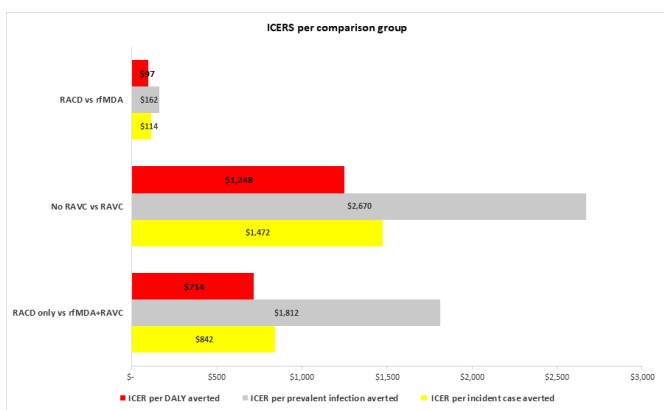

**Figure 3** ICER of RACD vs rfMDA, no RAVC vs RAVC and RACD only vs rfMDA+RAVC per incident case averted, prevalent infection averted and DALY averted. DALY, disability-adjusted life years; ICER, incremental cost effectiveness ratio; RACD, reactive case detection; RAVC, reactive focal vector control; rfMDA, reactive focal mass drug administration.

tested (for RACD), the total number of persons treated (for rfMDA) or the total number of persons residing in households sprayed (for RAVC) and (4) cost per PAR, calculated as the total cost of the intervention divided by the total population in the study arm in which the specific intervention was implemented.

For the cost of rfMDA and RAVC individually, we report costs relative to the control group because half of the clusters in these groups received a different intervention than the one being evaluated (ie, half of the EAs in the rfMDA group received RAVC and in the RAVC group half of the EAs received RACD and the other half received rfMDA), resulting in challenges to assess the cost and cost effectiveness of actual individual interventions. For the cost of rfMDA+RAVC combined, we report exact figures because

the analyses are restricted to EAs receiving these specific interventions (online supplemental appendix 2).

For cost effectiveness, the incremental cost effectiveness ratio (ICER) was calculated as the difference in costs between intervention and control divided by the difference in their effect: ICER=((cost of intervention)−(cost of control))/(effect of intervention−effect of control). For the effect, we used three different impact measures: incidence rates (ICER per incident case averted), infection prevalence (ICER per prevalent infection averted) and disability adjusted life years (DALYs) (ICER per DALY averted). Models used to estimate adjusted incidence rate ratios and adjusted prevalence ratios with 95% CIs, as reported in,[13] were used to predict incidence or infection prevalence. Using the population of the study area, the number of incident cases and prevalent infections was then estimated and used to calculate ICERs. DALYs averted were estimated using a mean loss of 1.18 DALYs per malaria case[26] with disability weights and life expectancy of 63.5 years for females and 58.9 years for males from the WHO life tables,[27] and an average duration of 7 days for a malaria episode without age weighting and discounting.[28]

To define the interventions as cost effective, we used Namibia's 2017 gross domestic product (GDP) per capita[29] and the WHO economic evaluation guidelines on cost-effectiveness thresholds for low-income and middle-income countries. The guidelines consider any programme costs that are less than the national per capita GDP per DALY averted to be highly cost effective. Programme costs that are less than three times the national per capita GDP per DALY averted are considered cost effective.[30]

To enable comparisons between study arms, personnel time is presented as time to complete an intervention event (from arrival at index case household to departure

**Table 4** Results of sensitivity analysis and scenario analyses (cost in 2017 in US$)

| Parameter | Base value/sensitivity analysis value(s) | rfMDA | | RAVC | | rfMDA+RAVC | |
|---|---|---|---|---|---|---|---|
| | | Costs per PAR | ICER (vs RACD) per incident case averted | Costs per PAR | ICER (vs No RAVC) per incident case averted | Costs per PAR | ICER (vs RACD only) per incident case averted |
| Reference | Predicted incidence*, Discount rate 3%, Personnel salaries on partner scale, Actellic 300 CS cost $38.80 | $41.4 | $114 | $48.8 | $1472 | $54.4 | $842 |
| Intervention effectiveness* | Lower 95% CI | NA | $63 | NA | $996 | NA | $625 |
| | Upper 95% CI | NA | $679 | NA | $2781 | NA | $1304 |
| Discount rate | 0% | $41.2 | $109 | $48.6 | $1464 | $54.1 | $840 |
| | 5% | $41.5 | $116 | $48.9 | $1477 | $54.5 | $851 |
| Personnel | Government scale | $32.0 | $86.1 | $37.9 | $1145 | $42.2 | $659 |
| | CHW | $35.2 | $110 | $40.4 | $1290 | $45.2 | $743 |
| Actellic 300 CS | $15 | NA | NA | $47.5 | $1381 | $52.9 | $799 |

Costs per PAR are NA for intervention effectiveness because cost inputs did not change.
Note that for personnel costs, government scale refers to use of the government pay scale for all staff. CHW refers to use of the CHW pay scale for nurses, data collectors and spray operators. Other positions (drivers and office staff) used a partner pay scale.
*95% CIs for predicted incidences (per 1000 person-years) are: 19 to 40 (rfMDA), 21.7 to 35.4 (RAVC) and 14.7 to 34.5 (rfMDA+RAVC).
CHW, community health worker; ICER, incremental cost effectiveness ratio; NA, not applicable; PAR, population at risk; RACD, reactive case detection; RAVC, reactive focal vector control; rfMDA, reactive focal mass drug administration.

**A**

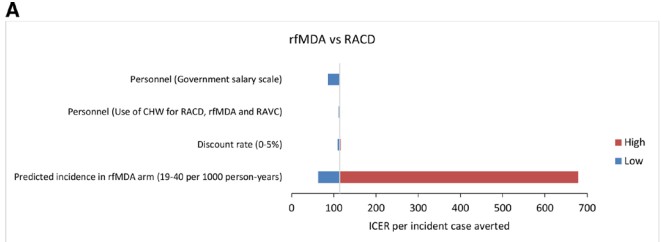

**B**

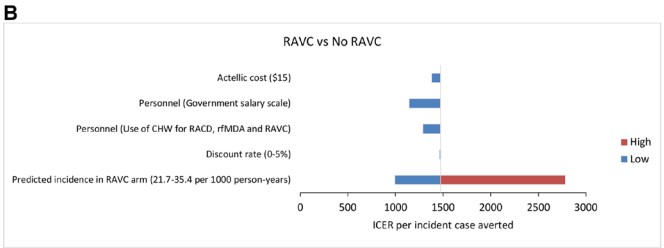

**C**

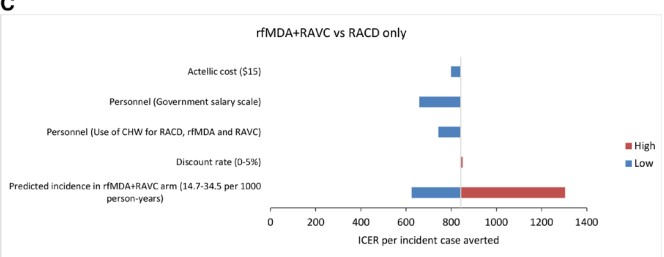

**Figure 4** Tornado diagram of change in the ICER per incident case averted for rfMDA vs RACD (A), RAVC vs no RAVC (B) and rfMDA+RAVC vs RACD only (C). The values in brackets are the range over which the parameter was varied. The vertical line is the baseline value of the ICER per incident case averted. The blue bars show the direction and magnitude of change in the ICER, when the input variable is set to its lower value and the red bars show the direction and magnitude of change when the input variable is set to its higher value. Marginal changes in the ICER are not visible on the graph. CHW, community health worker; DALY, disability-adjusted life years; ICER, incremental cost effectiveness ratio; RACD, reactive case detection; RAVC, reactive focal vector control; rfMDA, reactive focal mass drug administration.

from village of index case) excluding preparation time and driving time.

### Sensitivity analysis

A program evaluation and review technique (PERT) three-point estimation sensitivity analysis was performed to validate the cost-effectiveness outcome results. This analysis focused on varying the effectiveness of the interventions by varying malaria incidence using the lower and upper limit of the CI of predicted incidence in interventions arms while keeping median estimates unchanged for control arms.

The discount rate, personnel costs and consumables costs were also varied, and results obtained were compared with median estimates hereby referred to as best estimates. The discount rate was varied between 0% and 5%. For personnel, the governmental salary pay scale (online supplemental appendix 3) for all staff (nurses, spray operators, drivers, field supervisors, etc)

was used. Also, as malaria testing and treatment and IRS can be conducted by trained community health workers (CHWs), their compensation (standard rate of 260 NAD (US$18.20) as daily labour costs) was also considered. Finally, for the commodity price analysis, which was only used for RAVC, the government's subsidised price of $15 per bottle of pirimiphos-methyl (Actellic 300 CS) was considered.

### Patient and public involvement

Neither patients nor members of the public were involved in the design, conduct, reporting or dissemination plans of the study.

### RESULTS

During trial implementation, a total of 342 intervention events were conducted and 8948 individuals received RACD or rfMDA, and RAVC or no RAVC (mean: 26 participants/event). The total numbers of events conducted and individuals receiving each intervention by assigned group and by comparison arm are given in table 1.

The total cost of conducting rfMDA was 1.1× higher than RACD, while RAVC cost 1.7× more than no RAVC. The costs of rfMDA+RAVC were 1.8× higher than RACD only ($234 223 vs $127 312) (table 2). Personnel costs were the main cost driver for all three interventions, accounting for 66% of the RACD costs, 62% of rfMDA costs and 64% of RAVC costs (figure 1). As given in table 2 and figure 2, for all comparison groups, unit costs were higher in intervention groups compared with controls. The cost per PAR was 1.2× higher for rfMDA compared with RACD ($41.40 vs $35.80), 1.7× higher for RAVC compared with no RAVC ($48.80 vs $28.00) and 2× higher for rfMDA+RAVC than RACD only ($54.40 vs $26.90). In the RACD arms, there was a total of 114 infections identified, which translated to $3112 per infection identified.

Using incidence and prevalence outcome data from the trial (online supplemental appendices 1a and 1b), the number of malaria incident cases and prevalent infections averted among study arms population was estimated, and ICERs were calculated for each comparison group to evaluate cost effectiveness. The base case scenario results for impact and cost effectiveness are given in table 3. Compared with RACD only, rfMDA+RAVC had an ICER of $842 per incident case averted and $1812 per prevalent infection averted. The implementation of rfMDA resulted in an ICER of $114 per incident case averted and $162 per prevalent infection averted compared with RACD. RAVC was associated with an ICER of $1472 per incident case averted and $2670 per prevalent infection averted compared with no RAVC (figure 3).

Using mean DALYs loss estimates[28] and the number of incident cases averted, the implementation of rfMDA and RAVC translated into an ICER of $97 per DALY averted for rfMDA compared with RACD, $1248 per DALY averted for RAVC compared with no RAVC and $714 per DALY averted for rfMDA+RAVC compared with

**Table 5** Personnel time (min)

| | rfMDA | RACD | RAVC |
|---|---|---|---|
| Personnel-minutes per participant enrolled (median, IQR) | 30 (25–38) | 38 (35–44) | NA |
| Personnel-minutes per structure sprayed (median, IQR) | NA | NA | 34 (29–39) |
| Personnel minutes per individual protected with RAVC (median, IQR) | NA | NA | 9 (7–12) |
| Preparation time (min) (median, IQR) | 33 (27–39) | 32 (26–38) | 22.5 (17–28.5) |
| Travel time to and from community (min) | 124 (112–137) | 126 (113–140) | 138 (122–149) |

NA, not applicable; RACD, reactive case detection; RAVC, reactive focal vector control ; rfMDA, reactive focal mass drug administration.

RACD only (figure 3). Given that Namibia's 2017 GDP per capita was $6193,[29] both rfMDA and RAVC as implemented in Zambezi region, singly or combined, would be considered highly cost-effective health interventions.

### Sensitivity analysis

The parameters included in the sensitivity analysis and the change in costs and ICERs are detailed in table 4 and figure 4. For all three comparison groups, the ICER per incident case averted were sensitive to variations in predicted malaria incidence, and in personnel salary scale. Costs per PAR were sensitive to variations in personnel salary scale. Within the range of the variation used, discount rate and commodity cost (for Actellic 300 CS) did not have a significant impact on cost per PAR nor ICERs.

### Personnel time

Data regarding personnel time by intervention are given in table 5. The time spent by teams to enrol a participant was longer for RACD compared with rfMDA with a median personnel-minutes per participant enrolled of 38 min (IQR: 35–44 min) versus 30 min (IQR: 25–38 min). The median personnel-minutes per structure sprayed for RAVC was 34 min (IQR: 29–38 min). With an average of 4 people per household, the median personnel-minutes per individual protected with RAVC was 9 min. The median preparation time was 33 min for rfMDA and 32 min for RACD and RAVC. For all interventions, the teams spent on average 2 hours per event travelling to and from the index case household.

### DISCUSSION

Results of this study show that compared with RACD, reactive focal interventions (rfMDA, RAVC and rfMDA+RAVC combined) used in a low transmission Sub-Saharan African setting are cost effective, when considering outcomes of malaria incidence or prevalence, or DALYs averted. In addition, rfMDA was time saving compared with RACD. These results, taken together with the main trial findings on effectiveness of these interventions, suggest that in such settings, a change in policy and practice from RACD to rfMDA and/or RAVC should be considered.[11]

It is established that the investment needed to achieve malaria elimination will be higher per case than that of standard malaria control.[31 32] For example, the costs per person reported in this study for RAVC ($48.80) are significantly higher than the ones reported by White *et al* in a systematic review on costs and cost effectiveness of malaria control interventions ($2.20 for mosquito nets and $6.70 for IRS).[33] However, in the elimination context, a comparison to the current standard of care, which often includes RACD, is more useful. Furthermore, it is generally accepted that the long-term benefits of elimination outweigh the costs, making the pursuit worthwhile.[31 32]

Although rfMDA costs are marginally higher compared with RACD, the difference is mostly a reflection of economies of scale. Due to the way in which we have calculated costs in relation to events, we show a conservative assessment of costs and cost effectiveness. For example, because RACD was less effective, more RACD events were carried out and more people tested, lowering the cost per person. In the only other reported study of costs for RACD, conducted by Zelman *et al* in Indonesia,[34] the cost per individual screened, without major start-up capital costs, was lower ($28 compared with $39 in this study) but that study had more individuals enrolled per RACD event (42 individuals compared with 25 individuals per event in this study). Also, in this study, the total costs of the combined intervention and RACD only should not be compared with those of the individual interventions because the combination intervention was only conducted in half the study clusters.

In the sensitivity analyses, both rfMDA and RAVC implemented alone or in combination remained cost effective with varying effectiveness. The sensitivity analysis also provided an indication on how the costs of interventions may be reduced in the future. Costs could be reduced by more than 10% if the interventions are to be implemented by the ministry of health using existing resources (eg, personnel, vehicles, fuel, etc) (table 4). Costs could be further reduced through better integration of staff with other disease control programmes. Consistent with other studies of malaria community level interventions,[35 36] personnel costs were the main cost driver. Conducting reactive interventions is a challenging and resource intensive task, requiring dedicated teams working throughout the malaria transmission season to achieve high coverage of interventions. A CHW programme could be considered as a potentially cost-effective approach for the delivery of health services to communities.[37] CHWs are used in similar settings to deliver reactive interventions,[16 38 39] and Namibia currently has a pool of trained and accredited CHWs to test and treat malaria who could be used in the

implementation of reactive interventions. A community-based, self-administered treatment approach using village health workers is currently being tested in a randomised controlled trial in the Gambia as a cost effective way to deliver rfMDA.[40] Finally, considerable cost savings can be made by combining rfMDA or RACD teams and activities with RAVC, which we did not do in the study due to protocol design.

Despite the time requirements associated with rfMDA, such as reviewing potential contraindications and adverse effects, and ensuring treatment compliance, the time required to implement rfMDA was less, due to the time it takes to conduct an RDT, wait for the results, share the result with the participant and then provide treatment if indicated. Not surprisingly, the median personnel time per participant was shorter in the rfMDA arms compared with RACD. The shorter time per participant in rfMDA is expected to result in a higher number of individuals covered by rfMDA in a programme implementation setting, potentially leading to a lower cost per person due to economies of scale.

This study was limited in several ways. The intervention was not designed to minimise costs and maximise cost effectiveness. For example, in rfMDA arms, more than the targeted number of 25 individuals could have been enrolled per event with the same staff working time and could have resulted in reduced costs per person enrolled. The study design did not allow direct comparison between actual single interventions, since half of the EAs in the RACD and the rfMDA comparison groups received RAVC, and for RAVC, half of the EAs received RACD and half received rfMDA. Although all efforts were made to exclude costs related to research activities, it was not possible to identify the costs related to some research activities such as time needed to obtain informed consent, conduct interviews and collect research related blood samples, and, therefore, these costs could not be excluded from the analysis. Programme implementation costs of these interventions are, therefore, likely to be lower. Moreover, the inclusion of the wider social and economic benefits of malaria prevention (lost wages, direct healthcare costs and transport costs to seek care), though beyond the scope of this work, would likely result in higher estimates of cost effectiveness. Finally, we limited our assessment to the test interventions used in the trial, and used RACD and/or no RAVC as a comparison. It is possible that comparisons with other approaches, including non-targeted interventions, and/or use of a different control (eg, no RACD), would also be cost effective.[41]

In conclusion, in the low transmission setting of Namibia, rfMDA and RAVC implemented as singly or in combination are highly cost-effective interventions that can substantially reduce malaria transmission. Compared with RACD, rfMDA was time-saving, and for the combination intervention, the additional cost and time requirements for RAVC could be minimised by having a single team implement both rfMDA and RAVC at the same time. Given the frequent occurrence of outbreaks in low transmission areas of southern Africa, applying the rfMDA±RAVC approach may be effective at bringing outbreaks rapidly under control. These results offer a good indication of the best value for money, though additional research is needed to provide information on other factors to guide decision-making such as affordability and budget impact. Beyond ICERs, the net benefits framework provides additional guidance to decision-makers for the choice of interventions using the probability of the intervention to achieve elimination of transmission.[42] Findings are likely to be generalisable to other low transmission settings where cases of malaria are highly clustered around index cases.

**Author affiliations**
[1]Malaria Elimination Initiative, Global Health Group, University of California San Francisco, San Francisco, California, USA
[2]JoDon Consulting Group LLC, Atlanta, Georgia, USA
[3]Department of Pediatrics, The University of Texas Southwestern Medical Center, Dallas, Texas, USA
[4]Multidisciplinary Research Centre, University of Namibia, Windhoek, Namibia
[5]Department of Infection Biology, London School of Hygiene & Tropical Medicine, London, UK
[6]Division of Prevention Science, University of California San Francisco, San Francisco, California, USA
[7]Ministry of Health and Social Services, Zambezi Region, Katima Mulilo, Namibia
[8]Ministry of Health and Social Services, Windhoek, Namibia
[9]Faculty of Health Sciences, School of Pathology, University of the Witwatersrand, Johannesburg, South Africa
[10]Department of Infectious Disease Epidemiology, London School of Hygiene & Tropical Medicine, London, UK

**Acknowledgements** The authors would like to express their gratitude to the national and regional Ministry of Health and Social Services, and the local traditional leadership for their support. The authors would also like to acknowledge Megan McElroy, Simataa Nyathi, Flavian Libita, Chaze Sibeya and Katie Fox for their support in the preparation and collection of field data.

**Contributors** MH, RG, and IK led the conception and study design. DM, JS, AB, HS and PU contributed to the study design. HN led the data collection with support from CS-G, VS, BZ, MT, LMP, LW, LS and AM. HN and MH performed data cleaning and data analyses. JN, BW and M-SKD supported data analysis. HN wrote the first draft of the manuscript. HN and MH wrote the final draft of the manuscript. MH acts as the guarantor. All authors read and approved the final manuscript.

**Funding** This work was supported by grants from the Novartis Foundation (grant number: A122666), the Bill and Melinda Gates Foundation (grant number: OPP1138299) and Horchow Family Fund (grant number: 5300375400 to MSH).

**Competing interests** None declared.

**Patient and public involvement** Patients and/or the public were not involved in the design, or conduct, or reporting, or dissemination plans of this research.

**Patient consent for publication** Not applicable.

**Ethics approval** The main trial received ethical approval from the Namibia Ministry of Health and Social Services and the institutional review boards of the University of Namibia, University of California San Francisco and London School of Medicine and Hygiene. Consent was not obtained from study staff being observed as it was considered part of study supervision.

**Provenance and peer review** Not commissioned; externally peer reviewed.

**Data availability statement** Data are available upon reasonable request. Data supporting the conclusions of this article are available upon request.

for any error and/or omissions arising from translation and adaptation or otherwise.

**ORCID iD**
Henry Ntuku http://orcid.org/0000-0001-7218-2710

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
