## [Reviewer comments · BMJ Open]

ARTICLE DETAILS

TITLE (PROVISIONAL)	Cost and cost-effectiveness of reactive case detection (RACD), reactive focal mass drug administration (rfMDA), and reactive focal vector control (RAVC) to reduce malaria in the low endemic setting of Namibia: an analysis alongside a 2x2 factorial design cluster randomised controlled trial
AUTHORS	Ntuku, Henry; Smith-Gueye, Cara; Scott, Valerie; Njau, Joseph; Whittemore, Brooke; Zelman, Brittany; Tambo, Munyaradzi; Prach, Lisa; Wu, Lindsey; Schrubbe, Leah; Kang Dufour, Mi-Suk; Mwilima, Agnes; Uusiku, Petrina; Sturrock, Hugh; Bennett, Adam; Smith, Jennifer; Kleinschmidt, Immo; Mumbengegwi, Davis; Gosling, Roly; Hsiang, Michelle

VERSION 1 – REVIEW

REVIEWER	Sahu, Maitreyi University of Washington
REVIEW RETURNED	20-Jun-2021

GENERAL COMMENTS	Overall comments:  - Ntuku and colleagues show that rfMDA and RAVC are not just effective (as shown in the parent study published in Lancet), but also cost-effective when compared with RACD alone. This is an important contribution, and accordingly this paper is suitable for publication pending the following revisions outlined below. - Parts of this paper are a bit choppy to read and not well contextualized with the literature - in particular the discussion section. This paper would benefit from (1) clearer description of methods, (2) further careful review of language and flow, and (3) careful consideration and comparison with the existing health economics literature related to malaria interventions. Some specific suggestions are attached. Specific comments: Introduction  - Pg 4, Line 41: The literature review does not seem up to date: for example, the comment “to our knowledge there are no published economic data on RAVC” should be updated to include a recent Lancet paper from Bath and colleagues comparing reactive IRS with standard IRS: ://pubmed.ncbi.nlm.nih.gov/33640068/ Methods/Results  - Pg 6, Lines 8-28: Thanks for the detailed description of the interventions
---

	 - Pg 5, Lines 30-53: Please provide additional justification -- the choice of comparison groups and reasoning behind these choices difficult to follow without reading the parent study. - The authors nicely present four outcomes: total cost, cost per event, cost per individual, and cost per population at risk. However, a lot of the results and figures/tables focus on the total cost which seems the most unclear and least programmatically useful value given that other districts beyond this study will differ greatly in terms of population size and number of cases.  o First, can you clarify what the total cost represents – is it an annual cost or the cost during the full trial period? How many years does this cover? o Second, which cost do you think is the most informative and comparable? I would think this is the “cost per population at risk”? If so, can you focus on this in your primary results (including the multiplicative comparison) and include this in your Table 4 instead of total cost. - Some description of the DALY calculation needs to be included in the methods. It’s only mentioned that DALYs are calculated “derived from a study conducted in a different country”. What does this mean; what disability weights are used? - Given that your largest costs come from personnel, it’s useful to describe exactly how this was calculated. How comparable are the salaries from the “private pay scale” used within this study to regular programmatic implementation? Can you show in the sensitivity analysis what happens if programmatic salaries are used? - I’m surprised that in the current sensitivity analysis, use of volunteers instead of nurses does not change the total costs by much. Can you describe the assumptions further? Is there no equivalent substitution of cadre for RAVC? - Other than this, the authors have for the most part appropriately identified a few key parameters to vary in their sensitivity analysis, and in the absence of conducting any probabilistic sensitivity analysis have appropriately described their methods.  o In particular varying the predicted incidence seems important – but can you list the values for the predicted incidence which you varied (i.e. the actual values for the incidence IQRs) in an appendix table which you reference in your methods? o I am not convinced that showing the exclusion of startup costs is a useful sensitivity analysis, given that any program will have some startup o The sensitivity analysis figure requires clarification and better labels – what is the x-axis? The ICER? - Results on personnel time: this would be better shown in a figure or table (even if included in the supplement), in addition to the text Discussion  - RACD, rfMDA and RAVC are not the only possible interventions, so some additional comparison with other options including non-targeted interventions could be useful for policymakers. In addition, it’s useful to tie in with the broader health economic literature on this topic, for
--	--

	example: costing of non-targeted MDA and IRS: https://malariajournal.biomedcentral.com/articles/10.1186/s12936-020-03405-3  - An important limitation of this study to discuss is that no attempt is made to compare with the standard of care, which makes it difficult to compare the results from study with other similar ones despite the use of metrics which should be comparable, such as ICER per case/DALY averted. The reference is RACD, which is not recommended by the WHO given the current state of diagnostics – though as the authors mention it's often used in low transmission settings - A potential point for discussion: In general, cost-effectiveness is lower in a low endemic or elimination context. Antillon and colleagues define an additional premium of elimination for interventions which have higher likelihood of achieving elimination, which might be useful to tie in: https://www.medrxiv.org/content/10.1101/2021.02.10.20181974v1.full.pdf Minor comments: Abstract, Line 7: “Reactive focal mass drug administration”, not reactive focal drug mass administration Pg 12, Line 40: Please specify the commodity price analysis is only relevant for RAVC; and what the comparison price is Figures 1 and 3: please add colors because it's difficult to see the grey shading Figure 2: Please spell out “Population at risk (PAR)” Table 2 / Figures: It's easier for the reader to follow if these tables/figures match the order that you introduced them in the methods section (i.e. (1) total cost, (2) cost per event, (3) cost per individual, and (4) cost per population at risk ; also (1) RACD vs rfMDA, (2) RAVC v no RAVC,(3) rfMDA + RAVC vs RACD only) Pg. 6, Lines 38-39: Please clarify, this sentence is hard to follow given that rfMDA and RAVC are a combined intervention, as well as two separate interventions
--	---

VERSION 1 – AUTHOR RESPONSE

1. Overall comments:

Ntuku and colleagues show that rfMDA and RAVC are not just effective (as shown in the parent study published in Lancet), but also cost-effective when compared with RACD alone. This is an important contribution, and accordingly this paper is suitable for publication pending the following revisions outlined below.

Response: We agree with this overall conclusion regarding study findings, and appreciate the positive review.

2. Parts of this paper are a bit choppy to read and not well contextualized with the literature - in particular the discussion section. This paper would benefit from (1) clearer description of methods, (2) further careful review of language and flow, and (3) careful consideration and comparison with the existing health economics literature related to malaria interventions. Some specific suggestions are attached.

Response: We have revised the methods and discussion, with special attention to language and flow. See below regarding comparison with existing health economics literature related to malaria interventions.

3. Specific comments:

Introduction

- Pg 4, Line 41: The literature review does not seem up to date: for example, the comment “to our knowledge there are no published economic data on RAVC” should be updated to include a recent Lancet paper from Bath and colleagues comparing reactive IRS with standard IRS: [://pubmed.ncbi.nlm.nih.gov/33640068/](https://pubmed.ncbi.nlm.nih.gov/33640068/)

Response: Thank you for this comment. We now cite the Bath paper and updated this text as follows on page 4, line 18:

“Compared to standard IRS, RAVC has been shown to be cost-effective in a very low transmission setting [18]. To our knowledge there are no published economic data on RAVC, nor rfMDA+RAVC, in a low transmission setting.”

4. Methods/Results

- Pg 6, Lines 8-28: Thanks for the detailed description of the interventions

Response: Noted thank you.

5. Pg 5, Lines 30-53: Please provide additional justification -- the choice of comparison groups and reasoning behind these choices difficult to follow without reading the parent study.

Response: We have updated this paragraph with clarification regarding the rationale for this design on page 4, line 23:

“The goal of the trial was to investigate the effectiveness and feasibility of novel interventions that target the parasite reservoir in humans (rfMDA) and the vector (RAVC) and are applied in a reactive and focal fashion (e.g. in response to confirmed, passively identified malaria cases). These new interventions were compared singly and in combination to their respective current standard of care (e.g. rfMDA vs RACD, RAVC vs no RAVC, and rfMDA+RAVC vs RACD+no RAVC).”

6. The authors nicely present four outcomes: total cost, cost per event, cost per individual, and cost per population at risk. However, a lot of the results and figures/tables focus on the total cost which seems the most unclear and least programmatically useful value given that other districts beyond this study will differ greatly in terms of population size and number of cases. o First, can you clarify what the total cost represents – is it an annual cost or the cost during the full trial period? How many years does this cover?

Response: Total costs represent the costs for the one year of trial implementation. In the Data Analysis section, page 9, line 3, we clarify: “(1) total cost of the intervention for the duration of the one-year trial;”

7. Second, which cost do you think is the most informative and comparable? I would think this is the “cost per population at risk”? If so, can you focus on this in your primary results (including the multiplicative comparison) and include this in your Table 4 instead of total cost.

Response: We agree with the reviewer that cost per population at risk (PAR) is the most informative and comparable. We have included the cost per PAR in the multiplicative comparison (page 11, lines 17-20) “The cost per population at risk was 1.2x higher for rfMDA compared to RACD (\$41.4 versus \$35.8), 1.7x higher for RAVC compared to no RAVC (\$48.8 versus \$28.0) and two times higher for rfMDA+RAVC than RACD only (\$54.4 versus \$26.9)”. We have updated Table 4 to show Costs per PAR. See Table 4.

8. Some description of the DALY calculation needs to be included in the methods. It’s only mentioned that DALYs are calculated “derived from a study conducted in a different country”. What does this mean; what disability weights are used?

Response: We clarify the methods for this analysis on page 10, line 1-4: “DALYs averted were estimated using a mean loss of 1.18 DALYs per malaria case [26] with disability weights and life expectancy of 63.5 years for females and 58.9 years for males from the WHO life tables [27], and an average duration of 7 days for a malaria episode without age weighting and discounting [28].” The prior text regarding the calculations from a different country is not accurate. We used a similar method used in a nearby country, but the assumptions for DALY calculations are based on the Global Burden of Disease Study and we have cited this reference (reference 26: Murray CJL, Vos T, Lozano R, et al. Disability-adjusted life years (DALYs) for 291 diseases and injuries in 21 regions, 1990-2010: a systematic analysis for the Global Burden of Disease Study 2010. *Lancet*, 2012).

9. Given that your largest costs come from personnel, it’s useful to describe exactly how this was calculated. How comparable are the salaries from the “private pay scale” used within this study to regular programmatic implementation? Can you show in the sensitivity analysis what happens if programmatic salaries are used?

Response: Thank you for this suggestion. We have specified how personnel costs were calculated on page 7, lines 15-17, “Based on the time and motion observation (see below), a percentage of the personnel time spent on each intervention was applied to the full annual costs (including salary, benefits, and any types of allowances or per diems)”. We have also added a sensitivity analysis using lower governmental salaries. Note that we now refer to the research staff salaries as “partner pay scale” instead of “private pay scale.” The use of governmental vs partner pay scale reduced cost per PAR and ICERs for all comparisons (Table 4).

10. I’m surprised that in the current sensitivity analysis, use of volunteers instead of nurses does not change the total costs by much. Can you describe the assumptions further? Is there no equivalent substitution of cadre for RAVC?

Response: Community Health Workers (CHW) are not volunteers. In the Methods we provide their daily compensation rate. Using CHW, vs the study staff that were paid at partner pay scales, the costs are lower by ~10 to 20% for each of the interventions. CHW can also be trained to conduct RAVC. We have clarified this in the Methods, page 10, lines 22-24: “Also, as malaria testing and treatment and IRS can be conducted by trained community health workers (CHW), their compensation (standard daily rate of 260 NAD as daily labor costs) was also considered.”

11. Other than this, the authors have for the most part appropriately identified a few key parameters to vary in their sensitivity analysis, and in the absence of conducting any probabilistic sensitivity analysis have appropriately described their methods.
o In particular varying the predicted incidence seems important – but can you list the values for the predicted incidence which you varied (i.e. the actual values for the incidence IQRs) in an appendix table which you reference in your methods?

Response: This is a good suggestion. The values for the predicted incidences have been added to Table 4, and of note, they are also in Figure 4.

12. I am not convinced that showing the exclusion of startup costs is a useful sensitivity analysis, given that any program will have some startup

Response: Because programs may have different start-up costs (e.g. fewer if they already have teams conducting some sort of surveillance and response, and higher if they have no existing infrastructure), we have elected to keep the breakdown that shows startup with recurrent costs, and recurrent costs alone, and removed exclusion of startup costs in the sensitivity analysis.

13. The sensitivity analysis figure requires clarification and better labels – what is the x-axis? The ICER?

Response: The x-axis shows ICER per incident case averted and we have labeled the axes as such in Figure 4.

14. Results on personnel time: this would be better shown in a figure or table (even if included in

the supplement), in addition to the text

Response: We agree that this would be easier to digest in a figure or table. We have added a table (Table 5).

15. Discussion

- RACD, rMDA and RAVC are not the only possible interventions, so some additional comparison with other options including non-targeted interventions could be useful for policymakers. In addition, it's useful to tie in with the broader health economic literature on this topic, for example: costing of non-targeted MDA and IRS:

<https://malariajournal.biomedcentral.com/articles/10.1186/s12936-020-03405-3>

16. An important limitation of this study to discuss is that no attempt is made to compare with the standard of care, which makes it difficult to compare the results from study with other similar ones despite the use of metrics which should be comparable, such as ICER per case/DALY averted. The reference is RACD, which is not recommended by the WHO given the current state of diagnostics – though as the authors mention it's often used in low transmission settings

Response to Comment 15 and 16: We have added these important points, and the citation, in the Discussion, page 15, line 18-21: “Finally, we limited our assessment to the test interventions used in the trial, and used RACD and/or no RAVC as control. It is possible that comparisons with other approaches, including non-targeted interventions, and/or use a different control (e.g. no RACD), would also be cost-effective [40].” Indeed, WHO recommends against RACD (WHO Evidence Review Group on mass drug administration, mass screening and treatment and focal screening and treatment for malaria, 2015) as a strategy for transmission reduction. However, WHO does recommend RACD for surveillance (A framework for malaria elimination, 2017). Thus, similar to many low endemic settings, RACD is standard practice.

17. A potential point for discussion: In general, cost-effectiveness is lower in a low endemic or elimination context. Antillon and colleagues define an additional premium of elimination for interventions which have higher likelihood of achieving elimination, which might be useful to tie in: <https://www.medrxiv.org/content/10.1101/2021.02.10.20181974v1.full.pdf>

Response: Thank you for this helpful reference. We have added it to the Discussion page 16, lines 6-8: “Beyond ICERs, the net benefits framework provides additional guidance to decision makers for the choice of interventions using the probability of the intervention to achieve elimination of transmission.” <https://www.medrxiv.org/content/10.1101/2021.02.10.20181974v1.full.pdf>

18. Minor comments:

Abstract, Line 7: “Reactive focal mass drug administration”, not reactive focal drug mass administration

Response: Thank you. We have made the edit in the abstract and the title.

19. Pg 12, Line 40: Please specify the commodity price analysis is only relevant for RAVC; and what the comparison price is

Response: We have clarified in the methods page 10, line 24: “Finally, for the commodity price analysis, which was only used for RAVC, the government's subsidized price of \$15 per bottle of pirimiphos-methyl (Actellic 300 CS) was considered.”

20. Figures 1 and 3: please add colors because it's difficult to see the grey shading

Response: Colors have been added

21. Figure 2: Please spell out “Population at risk (PAR)”

Response: We have spelled out “population as risk.”

22. Table 2 / Figures: It’s easier for the reader to follow if these tables/figures match the order that you introduced them in the methods section (i.e. (1) total cost, (2) cost per event, (3) cost per individual, and (4) cost per population at risk ; also (1) RACD vs rfMDA, (2) RAVC v no RAVC,(3) rfMDA + RAVC vs RACD only)

Response: The order has been changed a suggested. See table 2.

23. Pg. 6, Lines 38-39: Please clarify, this sentence is hard to follow given that rfMDA and RAVC are a combined intervention, as well as two separate interventions

Response: We have updated this paragraph, page 6, lines 14-19, and refer to rfMDA+RAVC as “the combination.” We feel this edit makes the text easier to follow:

“The results of the trial showed that rfMDA and RAVC implemented alone and in combination reduced malaria incidence and parasite prevalence. Specifically, rfMDA and RAVC each reduced malaria incidence by nearly 50% when compared with their respective controls RACD and No RAVC, and their combination reduced malaria incidence by 75% compared to RACD only. Similarly, rfMDA and RAVC reduced prevalence by 41% and 64%, compared to their respective controls RACD and No RAVC and their combination reduced prevalence by 84% compared to RACD only [12].”

VERSION 2 – REVIEW

REVIEWER	Sahu, Maitreyi University of Washington
REVIEW RETURNED	26-Apr-2022
GENERAL COMMENTS	Thank you for this clear response! No additional comments from me.